# Lutein Can Alleviate Oxidative Stress, Inflammation, and Apoptosis Induced by Excessive Alcohol to Ameliorate Reproductive Damage in Male Rats

**DOI:** 10.3390/nu14122385

**Published:** 2022-06-09

**Authors:** Yebing Zhang, Haoyue Ding, Lei Xu, Suli Zhao, Shouna Hu, Aiguo Ma, Yan Ma

**Affiliations:** 1Institute of Nutrition and Health, School of Public Health, Qingdao University, Qingdao 266021, China; 18469112314@163.com (Y.Z.); qq604378593@163.com (H.D.); zsl199609@163.com (S.Z.); hushouna@163.com (S.H.); magfood@qdu.edu.cn (A.M.); 2Huai’an Center for Disease Control and Prevention, Huai’an 223003, China; xulei951121@163.com

**Keywords:** lutein, alcohol, testis, inflammation, oxidative stress, apoptosis

## Abstract

Chronic excessive alcohol intake may lead to male reproductive damage. Lutein is a carotenoid compound with antioxidant activity. The purpose of this study was to observe the effect of lutein supplementation on male reproductive damage caused by excessive alcohol intake. In this study, an animal model of excessive drinking (12 mL/(kg.bw.d)) for 12 weeks was established and supplemented with different doses of lutein (12, 24, 48 mg/(kg.bw.d)). The results showed that the body weight, sperm quality, sex hormones (FSH, testosterone), and antioxidant markers (GSH-Px) decreased significantly, while MDA and inflammatory factors (IL-6, TNF-α) increased significantly in the alcohol model group when compared to the normal control group. After 12 weeks of high-dose lutein supplementation with 48mg/(kg.bw.d), the spermatogenic ability, testosterone level, and the activity of marker enzymes reflecting testicular injury were improved. In addition, high-dose lutein supplementation downregulated the NF-κB and the pro-apoptosis biomarkers (Bax, Cytc and caspase-3), whereas it upregulated the expression of Nrf2/HO-1 and the anti-apoptotic molecule Bcl-2. These findings were fully supported by analyzing the testicular histopathology and by measuring germ cell apoptosis. In conclusion, lutein protects against reproductive injury induced by excessive alcohol through its antioxidant, anti-inflammatory, and anti-apoptotic properties.

## 1. Introduction

The incidence of infertility has increased in recent years. Among all infertile couples, the proportion of infertility caused by male factors accounts for about 50% [1]. A foreign study found [2] that male infertility is mainly due to the decrease in the number and motility of sperm in semen and the significant increase in the incidence of testicular diseases. The factors leading to the decline of sperm quality and testicular abnormalities include environmental pollutants, drugs, alcoholism, and smoking [1]. Among them, alcohol is the most common dietary factor that people are exposed to in daily life, which can induce serious male reproductive damage [3]. At present, the specific mechanism of alcohol-induced damage to testicular function has not been clarified, but the free radicals produced in the process of alcohol metabolism and oxidative stress are considered to play a key role in its toxicity [4]. Oxidative stress is caused by an imbalance in antioxidant responses in the body, which can lead to adverse reactions in tissues and cells, such as lipid peroxidation and DNA damage, and further induce inflammation and apoptosis [5]. However, drugs for preventing and alleviating alcoholism are scarce and have serious side effects [6]. The supplementation of natural active substances in a reasonable diet to reduce the toxicity caused by alcohol has become a direction worth exploring.

Lutein is one of the most widely distributed carotenoids in fruits and vegetables, and it mainly exists in dark green leafy vegetables and egg yolk. Animals obtain lutein directly or indirectly from the diet because the body cannot synthesize lutein [7]. Lutein is a component of macular pigment, so it is often used to protect the macula from photooxidation and enhance visual function [8]. In addition, lutein’s unique structure, containing conjugated double bonds and hydroxyl groups, can serve an antioxidant role through providing electrons with free radicals to produce more stable substances [9]. A study [10] on the intervention of lutein on ethanol-induced liver damage in rats indicated that the level of antioxidant enzymes in the livers of ethanol-treated rats was significantly increased with lutein treatment, thus reducing ethanol-induced hepatocyte injury. This further shows that lutein has a certain inhibitory effect on alcoholism-related toxicity; because the liver is the target organ of alcohol metabolism and can store a large amount of lutein, lutein’s protective effect in the liver may be more obvious than in other organs. However, there are few reports on whether lutein supplementation can alleviate the testicular damage caused by alcohol and whether the mechanism is the same as in previous studies.

Therefore, in this study, we established an animal model of reproductive damage caused by long-term (3 months) excessive drinking in male rats. In addition, different doses of lutein were supplemented to observe the effect of lutein on male reproductive damage caused by excessive alcohol. This study was designed to provide a reference for further revealing the harm of alcohol and the reduction of the damage with lutein supplementation.

## 2. Materials and Methods

### 2.1. Materials

Lutein was purchased from Yuanye biological company (Shanghai, China), with purity ≥80%. Alcohol (56% (*v*/*v*) ethanol) was purchased from Beijing Hongxing Alcohol Co., Ltd. (Beijing, China). Superoxide dismutase (SOD), malondialdehyde (MDA), glutathione peroxidase (GSH-Px), alkaline phosphatase (AKP), succinic dehydrogenase (SDH), and lactate dehydrogenase (LDH) assay kits were purchased from Nanjing Construction Co., Ltd. (Nanjing, China). Enzyme-linked immunosorbent assay (ELISA) kits for the detection of testosterone, luteinizing hormone (LH), estradiol (E2), and follicle-stimulating hormone (FSH) were purchased from Zhenke Biological Technology Co., Ltd. (Shanghai, China). Interleukin (IL)-1β, IL-6, and tumor necrosis factor α (TNF-α) kits were obtained from Wuhan Boshide Biological Engineering Co., Ltd. (Wuhan, China). Cell/tissue radioimmunoprecipitation (RIPA) lysis buffer and antibodies against β-actin were all obtained from Abways (Shanghai, China). The antibodies against Bcl-2, Bax, caspase-3, cleaved caspase-3, caspase-9, NF-κB (P65), the inhibitor of nuclear factor kappa B α (IκBα), Nrf2, and HO-1 were obtained from Abcam (Cambridge, England). Secondary antibodies and the bicinchoninic acid (BCA) protein assay kits were obtained from Bioeasy (Beijing, China) and Beyotime (Zhenjiang, Jiangsu, China), respectively.

### 2.2. Experimental Design and Animals

Seven-week-old male rats weighing between 140 and 160 g were purchased from Sperford Biotechnology Co., Ltd. (License No.: SCXK [Jing] 2019-0010). The rats were raised in a room (12 h light–dark cycle) with a temperature of 22 ± 1 °C and a humidity of 50 ± 5%, and were free to eat feed and drink water. All animal experiments strictly abided by the Guidelines for the Care and Use of Laboratory Animals of the National Institutes of Health, and were approved by the Animal Care and Use Committee of the Medical College (No.20210315Wistar900706) at Qingdao University.

Sixty rats were randomly separated into the following six groups (10 in each group): Normal control (CON), a normal diet with corn oil gavage equal to the other groups; lutein control (LUT), 24 mg/kg lutein (dissolved in corn oil); alcohol model group (ALC), 6, 8, 10 and 12 mL/(kg.bw.d) of 56% (*v*/*v*) ethanol was given in order to make rats adapt to alcohol intake. After the body weight of the rats was stable, 12 mL/(kg.bw.d) of 56% (*v/v*) ethanol and the same amount of corn oil were given every day. Low doses of lutein (12 mg/kg.bw) plus alcohol (ALC + LUT1), a medium dose of lutein (24 mg/kg.bw) plus alcohol (ALC + LUT2), and a high dose of lutein (48 mg/kg.bw) plus alcohol (ALC + LUT3)-treated groups gave the same alcohol intake as the ALC group. The whole experiment lasted for 12 weeks. At the end of the experiment, the rats were anesthetized, the blood was taken from the abdominal aorta, and the serum and plasma were collected and stored at −80 °C. Serum was used to detect the levels of serum testosterone, FSH, E2 and LH. The testis and epididymis of rats were used to detect histopathological and biochemical indexes.

### 2.3. Calculation of the Testis Index and Sperm Analysis

The calculation for the testis index is as follows: Testis index (%) = Testis weight(mg)/body weight(g) × 100. The left epididymis of the rats was placed in a 2 mL centrifuge tube with normal saline, and cut into pieces with ophthalmic scissors. Samples were bathed for 20 min in a water bath at 37 ℃. Part of the sperm suspension was taken, and the sperm count and sperm motility were determined by a blood cell meter [11]. An amount of 20 μL of sperm suspension was dripped on clean glass slides, dried in air, and fixed in methanol for 5 min. The dried slides were stained in 2% eosin for 1 h, and the sperm morphology was observed under a high-power microscope after drying. We counted five hundred sperm per tablet and calculated the rate of sperm deformity.

### 2.4. Histopathological Study of Testis Tissues

The rat testes were exposed to 4% paraformaldehyde solution for 48 h and dehydrated with graded alcohol (50–95%). After treatment with xylene, they was embedded in paraffin and cut into 4 μm thick slices. The sections were stained with hematoxylin–eosin and observed under the photoelectron microscope (Nikon, Tokyo, Japan).

### 2.5. Detection of Sex Hormones and Labeling Enzymes

Serum testosterone, FSH, and LH were measured using the commercial ELISA kits (Zhenke, Shanghai, China). The supernatant of each tissue homogenate obtained with centrifugation was used to determine SDH, LDH, ACP, and AKP in rat testicular tissue with colorimetric kits (Jiancheng, Nanjing, China).

### 2.6. Oxidative Stress Levels and Inflammatory Parameters

The levels of MDA, SOD and GSH-Px were determined to estimate the oxidative stress levels in rat testes using colorimetric kits according to the protocols recommended by the manufacturer (Jiancheng, Nanjing, China). The testis homogenate was used for the determination of the following testis inflammatory parameters: IL-1β, IL-6, and TNF-α. The levels of the inflammatory factors were determined using commercially available ELISA assays, following the instructions supplied by the manufacturer.

### 2.7. Western Blot Analysis

Homogenized testis tissue (50 mg) mixed with RIPA–phenylmethylsulphonyl fluoride (PMSF) (*v*/*v*, 1600:320 μL) was centrifuged at 14,000 rpm for 30 min. The total protein concentration was determined with a BCA Protein Assay kit. After incubating at 100 °C for 10 min, equal amounts of proteins (40 µg) were separated with 12% sodium dodecyl sulfate (SDS)–polyacrylamide gel electrophoresis (PAGE) and transferred onto polyvinylidene fluoride (PVDF) membranes (Merck Millipore). The membrane was then incubated in diluted Tris-Buffered Saline Tween (TBST) buffer with 10% skimmed milk at room temperature. After the membranes were washed with TBST, primary antibodies (Bax: 1:10,000; Bcl-2: 1:2000; Cas3: 1:6000; cleaved cas3: 1: 3000; Cas9: 1:3000; NF-κB(P65): 1:1000; IκBα: 1:1000; Nrf2: 1:1000; and HO-1: 1:1000) were added for incubation overnight at 4 °C. The corresponding secondary antibodies (dilution ratio 1:10,000) were then added to incubate at room temperature. After washing with TBST, the grayscale of the protein bands was analyzed.

### 2.8. TUNEL Staining

The apoptosis level of testicular cells was detected by TUNEL (terminal deoxynucleotidyl transferase terminal dUTP nick end labeling) staining on paraffin sections, and the steps were carried out according to the instructions of the kit. Images were observed using an ECLIPSE C1 fluorescence microscope (Nikon, Tokyo, Japan).

### 2.9. Statistical Analysis

The Statistical Product and Service software (SPSS, Chicago, IL, USA) was used to analyze data. A one-way analysis of variance (ANOVA) was used for multiple comparisons. Values are expressed as mean ± standard deviation. Values of *p* < 0.05 were considered statistically significant.

## 3. Results

### 3.1. Body Weights and Organ Weights

As shown in Figure 1, there was no difference in body weight among the groups before intervention. After 12 weeks of feeding, the weight, as well as the testicular and epididymis weight, of the ALC group were significantly lower than that of the CON group (*p* < 0.05). Testicular and epididymal weights and testicular and epididymal coefficients were significantly higher in the ALC + LUT2 group than those in the ALC group (*p* < 0.05). In the ALC + LUT3 group, body, testicular, and epididymal weights and the epididymal coefficient were significantly higher than those in the ALC group (*p* < 0.05). In addition, the testis and epididymis weights in the ALC + LUT1 group were significantly lower than those in the ALC + LUT2 and ALC + LUT3 groups, and the epididymal coefficient in the ALC + LUT1 group was significantly lower than that of the ALC + LUT2 group (*p* < 0.05).

### 3.2. Effect of Lutein on Sperm Quality in Alcohol-Treated Rats

As shown in Figure 2A, the morphology of the spermatozoa in the CON group was normal, the head was oval, and there was no obvious folding, breaking, or winding of the tail. When compared with the CON group, the abnormal morphology of spermatozoa in the ALC group was significantly increased, mainly characterized by broken tails, double tails, folding, and abnormal head morphology. When compared with the ALC group, the sperm morphology of rats in the ALC + LUT3 group was significantly improved and the number of abnormal sperm decreased. In Figure 2B–D, when compared with the CON group, the ALC group showed a decreased sperm count and motility by 31.11 and 53.2%, respectively. The decrease of sperm motility was statistically different (*p* < 0.05), but there was significant increase in sperm deformity rate by 2.7-fold (*p* < 0.05). In addition, the study also found that when compared with the CON group, the LUT group had a higher sperm count, motility, and lower sperm deformity rate, although there was no statistical difference.

However, the ALC + LUT2 group induced a significant increase in sperm count by 50.59% and reduced the rate of sperm deformity by 50.7% when compared with those of the ALC group (*p* < 0.01). The ALC + LUT3 group significantly increased sperm motility by 70.28% and reduced the rate of sperm deformity by 53.62% when compared with the ALC group (*p* < 0.05). In addition, when compared with the ALC + LUT3 group, the sperm deformity rate of the ALC + LUT1 group and ALC + LUT2 group was significantly higher than that of the high-dose group by 37.85 and 15.10%, respectively (*p* < 0.05), indicating that the supplement of high-dose lutein was more effective in improving the rate of sperm deformity.

### 3.3. Pathological Changes of Testicular Tissue

In the CON group and LUT group, the morphology and structure of testis were normal, the spermatogenic cells at all levels were normal, and the spermatogenic epithelium of the testis was arranged orderly and tightly (Figure 3). However, excessive alcohol reduced the number of germ cells in the lumen and caused degeneration of seminiferous tubules. At the same time, Sertoli cells fell off into the seminal duct, causing germ cell arrangement disorder and cell necrosis (Figure 3). After 12 weeks of lutein intervention, germ cell apoptosis in the ALC + LUT1, ALC + LUT2, and ALC + LUT3 groups was improved by varying degrees, and the reversal of injury was more pronounced in the ALC + LUT3 group. In the ALC + LUT3 group, the pathological damage of testicular tissue induced by alcohol was less, healthy spermatogenic cells could be observed, and there were a large number of spermatogenic cells and spermatozoa in the cavity (Figure 3).

### 3.4. The Levels of Marker Enzymes in Testis

There were significant decreases of the AKP, ACP, and SDH ratio in testicular tissues by 34.1, 20.72, and 21.9%, respectively, in the ALC group when compared with the CON group (*p* < 0.05). However, the ALC + LUT3 group showed a significant elevation in the testicular ratios of AKP and SDH, by 28.96 and 14.1%, respectively (*p* < 0.05), when compared with the ALC group, as shown in Table 1. In the LUT group, LDH tended to be higher than that of the CON group (*p* > 0.05). The activity of AKP in the ALC + LUT1 group was 32.13% lower than that in the ALC + LUT3 group, and the difference was statistically significant (*p* < 0.05).

### 3.5. Male Reproductive Hormone Level

As shown in Figure 4, serum testosterone and FSH levels in serum were all significantly lower in the ALC group than in the CON group (*p* < 0.05). However, serum testosterone and LH levels were significantly higher in the ALC + LUT3 group than in the ALC group (*p* < 0.05). The testosterone level of the ALC + LUT1 group was significantly lower than that of ALC + LUT3 group by 19.39% (*p* < 0.05).

### 3.6. Changes in Oxidative Stress Levels in Testicular Tissue

Alcohol significantly decreased the GSH-PX by 25.9% and increased the concentration of the lipid peroxidation product (MDA) by 36.3% in testicular tissues when compared with the CON group (*p* < 0.05). Rats in the ALC + LUT3 group had a significantly lower testicular level of MDA by 42.6% when compared with the ALC group (*p* < 0.05). Although the difference was not statistically significant, the concentrations of SOD and GSH-Px appeared to be higher in the ALC + LUT3 group than in the ALC group, as shown in Table 2. In the LUT group, the level of MDA was significantly lower than that in ALC group (*p* < 0.05).

### 3.7. Changes in the Levels of Inflammatory Markers in Testicular Tissue

The inflammatory parameters IL-6 (Figure 5A), IL-1β (Figure 5B), and TNF-α (Figure 5C) increased when rats were chronically exposed to alcohol; specifically, IL-6 and TNF-α were significantly different from the CON group (*p* < 0.05). When compared with those in the ALC group, IL-6 and TNF-α in the ALC + LUT3 group were significantly lower (*p* < 0.05). After 12 weeks of high-dose lutein intervention, alcohol-induced testicular inflammation in rats was significantly improved, indicating that lutein can reduce the occurrence of inflammation. The levels of IL-6 and TNF-α in the LUT group were significantly lower than those in the ALC group (*p* < 0.05), but there was no significant change when compared with those in the CON group.

### 3.8. Changes of Expression of Proteins Related to NF-κB/Nrf2 Pathway

Quantification of the levels of related protein factors in the testicular tissue via Western blot was performed, as shown in Figure 6. The administration of alcohol generated the downregulation of Nrf2, HO-1, and IκBα expression in rat testis tissues, concomitant with upregulated expression of NF-κB (P65). After 12 weeks of lutein supplementation, compared with CON group, Nrf2 and IκBα protein expressions were significantly upregulated in the ALC + LUT2 group. Nrf2, HO-1, and IκBα protein expression were significantly upregulated in the ALC + LUT3 group, while NF-κB (P65) was down-regulated to nearly normal levels. The expression of IκBɑ was also different in the various lutein dosage groups, and the expression of lutein in the low-dose lutein group was significantly lower than that in the high-dose group (*p* < 0.05).

### 3.9. Lutein Attenuates Chronic Alcohol-Induced Testicular Germ Cell Apoptosis

As shown in Figure 7, Bax, caspase-3, cleaved caspase-3, and caspase-9 were all upregulated in the ALC group (*p* < 0.05) when compared with the CON group. Caspase-3, cleaved caspase-3, and Bax were downregulated in the ACL + LUT3 group when compared with those in the ALC group, whereas Bcl-2 was notably upregulated when compared with that in the ALC group (*p* < 0.05). The expression levels of Bax, caspase-3, and cleaved caspase-3 in the ACL + LUT1 group and Bax and caspase-3 in the ACL + LUT2 group were significantly higher than those in the ACL + LUT3 group (*p* < 0.05). The expression level of cytochrome c (Cytc) in immunoblotting can reflect the role of mitochondria on testicular germ cells in the endogenous apoptosis pathway. As shown in Figure 7H, a pronounced elevation of Cytc was observed in the ALC group when compared with the CON group (*p* < 0.05). Cytc in the ALC + LUT2 and ALC + LUT3 groups were significantly lower than that of the ALC group (*p* < 0.05). These results suggested that Cytc was released from mitochondria to activate downstream apoptotic molecules. However, lutein inhibited the release of Cytc from mitochondria, especially in the ALC + LUT3 group (*p* < 0.05), and protected the testes from apoptosis induced by chronic alcohol ingestion, as reflected in the decreased TUNEL-positive cells (Figure 7I).

## 4. Discussion

In our study, the body and organ weights in rats in the alcohol model group decreased significantly after 12 weeks of an alcohol intervention, a finding that is consistent with a previous study [12]. The weight loss may be due to the disorder of appetite and gastrointestinal physiology, as well as impaired nutrient absorption due to the systemic toxicity of alcohol [13]. The decrease in testicular weight may be due to the decrease in the number of seminiferous tubule and germ cells, as well as the inhibition of spermatogenesis and steroidogenic enzyme activity [13]. With high-dose lutein (48 mg/kg.bw∙d) supplementation, body weight and reproductive organ development gradually returned to a normal level.

It is well known that long-term excessive drinking can lead to harm to some organs and tissues of the body, and the testes are more sensitive than other organs to alcohol toxicity [14]. In this study, the number and motility of sperm was reduced, the rate of sperm deformity increased, the epithelium of seminiferous tubules were exfoliated and atrophied in testicular tissue, germ cells were vacuolated, and lipid droplets formed after 12 weeks of alcohol intervention. These findings are consistent with previous research results [15]. The decline in sperm quality caused by excessive drinking may be affected by the activity of testicular marker enzymes and the levels of male reproductive hormones. Testicular marker enzymes can be used to evaluate the severity of testicular tissue injury, and also affect the formation and maturation of spermatozoa [16]. The marker enzyme ACP, located in Sertoli cells, scavenges damaged or senescent cells, and maintains the normal metabolism of testis. AKP is related to cell proliferation, metabolism, energy transfer, and other activities [16]. SDH is an important enzyme in the mitochondrial Kreb cycle, and is mainly related to the aerobic oxidation of acetyl-CoA and the production of ATP [17], affecting the development of germ cells [18]. In this study, the activities of the marker enzymes AKP, ACP, and SDH decreased significantly after prolonged, excessive drinking. Some studies have suggested that the disorder of marker enzymes caused by alcohol loading may be due to the strong cytotoxicity of mitochondrial lipid peroxidation in the testis, resulting in a disorder of energy metabolism, oxidative phosphorylation, Kreb cycle, and glycolysis. However, the disorder of marker enzymes can further affect the production of nutrients needed to support sperm maturation [19]. When compared with those in the normal control group, sex hormones (FSH and testosterone) in the alcohol model group were significantly decreased. FSH and testosterone are regulated by the hypothalamus–pituitary–stromal cells/gonadal axis [20]. The hypothalamus secretes gonadotropin-releasing hormone (GnRH), which stimulates the production of FSH in the pituitary gland. FSH stimulates the expression of androgen-binding protein (ABP) in Sertoli cells through binding to the FSH receptor [20]. ABP promotes spermatogenesis through transporting and protecting testosterone from degradation and maintaining the dynamic balance of testosterone in the testis [21]. With the secretion of GnRH, the secretion of estrogen (interstitial-cell-stimulating hormone, ISCH) in the interstitial cells of the adenohypophysis changes concomitantly. The arrival of ISCH in the blood to the testis can promote the secretion of testosterone by Leydig cells, whereas the testosterone in the blood can act on the hypothalamus and inhibit the secretion of GnRH. Through negative feedback regulation, the concentration of testosterone in the blood can be maintained in the normal range [22]. The findings above indicate that excessive drinking causes different aspects of damage to male reproductive function. However, after 12 weeks of lutein supplementation, these injuries were improved to some extent, and the sperm count and motility, marker enzymes (AKP and SDH), and sex hormones (e.g., testosterone) were significantly increased in the high-dose lutein group. The spermatogenic effect of lutein may be attributed to improving the performance of interstitial cells, inhibiting the degradation of testosterone, and regulating the levels of AKP and SDH to provide sufficient energy for sperm production.

Prior studies have suggested that ethanol-induced reproductive toxicity may be related to the excessive production of ROS and lipid peroxidation products [23]. MDA is a terminal metabolite produced by free radicals attacking cell membranes, a process known as lipid peroxidation. MDA levels can indirectly reflect the level of free radicals in testicular tissue and the degree of damage due to lipid peroxidation in testicular tissue [24]. Excessive ROS can cause damage to male reproductive health, including abnormal spermatogenesis, oligozoospermia, and decreased levels of reproductive hormones [25]. GSH-Px is the first line of defense against ROS. GSH-Px is a sulfur-containing antioxidant enzyme that promotes the reaction between reduced glutathione and hydrogen peroxide to produce oxidized glutathione and water [26]. It reduces the level of free radicals in the body by reducing hydrogen peroxide and alkane hydroperoxide and thereby protecting the structural and functional integrity of cell membranes [26]. In this study, excessive drinking significantly decreased the GSH-Px, increased MDA levels, and activated the oxidative stress response. The decrease of GSH-Px levels will hinder the normal growth of testicular tissue, including spermatogenic cells and Leydig cells, thus affecting the production of testosterone. MDA, which reflects the level of lipid oxidation, is closely related to the activity of marker enzymes. As an antioxidant nutrient, lutein’s protective effect on alcohol-induced reproductive toxicity may be related to its abilities of antioxidation and scavenging free radicals [27]. After supplementation with high-dose lutein (48 mg/kg∙d), the level of MDA in testis decreased significantly, whereas GSH-Px increased, which improved the oxidative stress induced by alcohol and maintained the normal testicular structure and sex hormone levels. In addition, this study also found that lutein supplementation can upregulate the expression of Nrf2/HO-1 protein; Nrf2 is a key transcription factor in the antioxidant defense system [28]. It protects cells and tissues from oxidative stress by regulating key signaling pathways such as those involving HO-1.

In this study, the prolonged exposure of rats to excessive alcohol resulted in an increase in pro-inflammatory molecules, namely IL-6 and TNF-α. TNF-α in the testis comes from spermatids, spermatocytes, and macrophages and is an important factor reflecting the health of germ cells [29]. TNF-α can aggravate body injury through increasing the release of oxygen free radicals and enhancing the effect of IL-6 [30]. However, rats in the high-dose lutein supplement group had lower levels of inflammatory cytokines (IL-6 and TNF-α) and a lower expression of NF-κB than those in the alcohol-treated group. NF-κB is the active center that regulates the cellular inflammatory response. The secretion of intracellular IL-6, IL-1β, and TNF-α is regulated by NF-κB [31,32]. Wanqiang [33] found that in vitro lutein inhibited neuroinflammation in lipopolysaccharide-activated BV-2 microglia through downregulating NF-κB levels.

Apoptosis is a programmed form of cell death controlled by several genes. Apoptosis of spermatogenic cells in testicular tissue directly affects the quality of sperm [34]. The pathway of mitochondrial-mediated apoptosis is mainly dominated by the Bcl2 protein family, which includes both anti-apoptosis (Bcl2) and pro-apoptosis (Bax) members [35]. The protein family controls the release of Cytc from the mitochondria, increases Cytc in the cytoplasm, binds with APAF-1 (activator of apoptotic enzyme-1) and caspase-9 to form an apoptotic body, further activates caspase-3 to cleave into cleaved caspase-3, and initiates the apoptotic pathway [36]. This study found that excessive drinking can increase the permeability of the mitochondrial membrane, leading to the release of Cytc, and stimulating the production of caspase-3. Mohammad [37] found that ethanol could activate the apoptosis signal pathway of the gastric mucosa, resulting in a significant upregulation of the pro-apoptotic proteins Bax and caspase-3 and a significant downregulation of gastric mucosal Bcl2 activity. These findings are consistent with our research results. In this study, lutein treatment inhibited the expression of Bax, prevented the release of Cytc from mitochondria, and inhibited the malignant apoptosis cascade induced by ethanol. To further observe the apoptosis of germ cells in rats, we carried out fluorescent TUNEL experiments on testicular tissue. The results showed that the number of apoptotic testicular germ cells was larger in the alcohol model group, and most were late apoptotic cells. In the lutein treatment group, the apoptotic cells decreased gradually with the increase of lutein dose, and the effect was more obvious in the high dose group. Therefore, our results show that lutein has anti-apoptotic effects because of its antioxidant capacity and a protective effect on mitochondria.

## 5. Conclusions

As far as is known, this is the first study exploring the protective effect of lutein against male reproductive damage caused by chronic alcohol poisoning. Long-term excessive drinking will cause obvious oxidative stress in the testicular tissue and damage the spermatogenic ability and testicular function of men. Lutein protects against such damage, with high doses of lutein having the best effect. The protective effect of lutein is mediated by inhibiting lipid oxidation (MDA formation) and inflammatory factors (IL-6 and TNF-α), supporting the endogenous antioxidant defense system of testicular tissues and decreasing morphological and histopathological injury. Lutein supplementation can be considered as an adjuvant therapy for the prevention of reproductive dysfunction caused by chronic alcohol poisoning.

## Figures and Tables

**Figure 1 nutrients-14-02385-f001:**
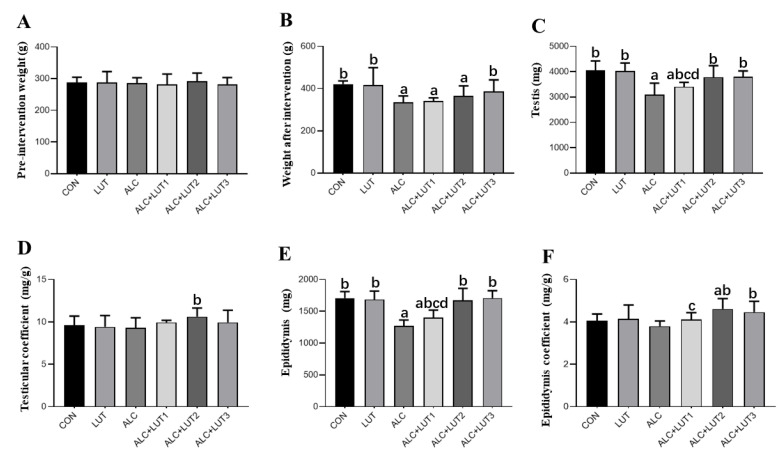
Body weights and organ weights. (**A**) Pre-intervention weight; (**B**) Weight after intervention; (**C**) Testicular weight; (**D**) Testicular coefficient; (**E**) Epididymis weight; (**F**) Epididymis coefficient. CON: control group; LUT: Lutein alone supplement group; ALC: alcohol model group; ALC + LUT1: alcohol + low-dose Lutein group; ALC + LUT2: alcohol + medium-dose Lutein group; ALC + LUT3: alcohol + high-dose Lutein group; a, vs. the CON group, *p* < 0.05; b, vs. the ALC group, *p* < 0.05; c, vs. the ALC + LUT2 group, *p* < 0.05, d, vs. the ALC + LUT3 group, *p* < 0.05 (n = 10, mean ± SD).

**Figure 2 nutrients-14-02385-f002:**
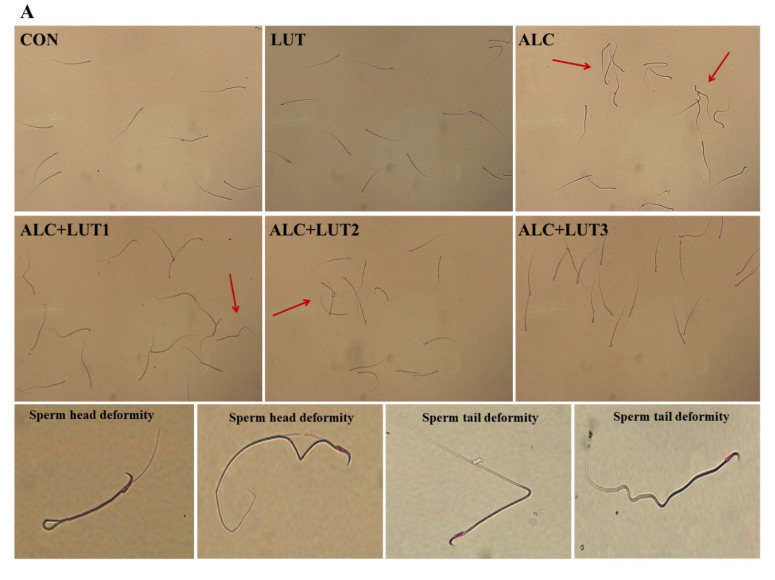
Effect of lutein on sperm quality in alcohol-treated rats. (**A**) Sperm morphology; (**B**) sperm count; (**C**) sperm motility; (**D**) sperm abnormality; CON: control group; LUT: Lutein alone supplement group; ALC: alcohol model group; ALC + LUT1: alcohol + low-dose lutein group; ALC + LUT2: alcohol + medium-dose lutein group; ALC + LUT3: alcohol + high-dose lutein group; The red arrow is the malformed sperm; a, vs. the control group, *p* < 0.05 (CON); b, vs. the alcohol group, *p* < 0.05 (ALC); c, vs. the ALC + LUT2 group, *p* < 0.05, d, vs. the ALC + LUT3 group, *p* < 0.05 (n = 10, mean ± SD).

**Figure 3 nutrients-14-02385-f003:**
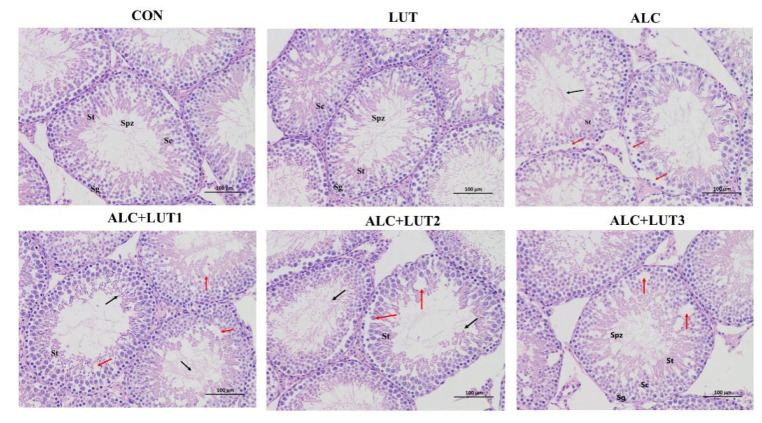
Morphological changes in testis tissue. CON: control group; LUT: lutein alone supplement group; ALC: alcohol model group; ALC + LUT1: alcohol + low-dose lutein group; ALC + LUT2: alcohol + medium-dose lutein group; ALC + LUT3: alcohol + high-dose lutein group; Sg (spermatogonia), ST (seminiferous tubule), SC (Sertoli cell), SPZ (sperm); black arrow (alcohol causes convoluted seminiferous tubule degeneration, causing Sertoli cells to fall off into the lumen of the convoluted seminiferous tubule), red arrow (seminiferous epithelium showing severe seminiferous tubule atrophy, including germ cell disorder and necrotic area). Sections were stained with hematoxylin and eosin (100×).

**Figure 4 nutrients-14-02385-f004:**
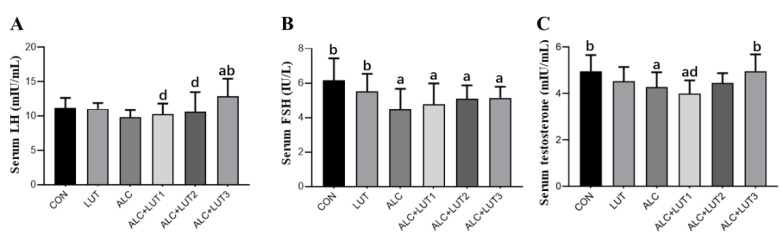
Male reproductive hormone levels. (**A**) Luteinizing (LH) hormone levels; (**B**) Follicle-stimulating hormone (FSH) levels; (**C**) Testosterone levels; CON: control group; LUT: lutein alone supplement group; ALC: alcohol model group; ALC + LUT1: alcohol + low-dose lutein group; ALC + LUT2: alcohol + medium-dose lutein group; ALC + LUT3: alcohol + high-dose lutein group; a, vs. the control group, *p* < 0.05 (CON); b, vs. the alcohol group, *p* < 0.05 (ALC); d, vs. the ALC + LUT3 group, *p* < 0.05. (n = 10, mean ± SD).

**Figure 5 nutrients-14-02385-f005:**
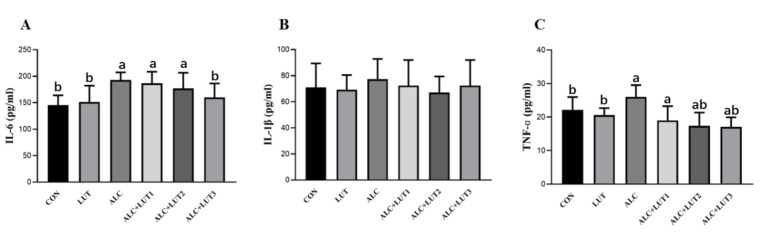
Changes in levels of inflammatory markers in testicular tissue. (**A**) Interleukin (IL)-6 levels; (**B**) IL-1β levels; (**C**) tumor necrosis factor α (TNF-α) levels. CON: control group; LUT: lutein alone supplement group; ALC: alcohol model group; ALC + LUT1: alcohol + low-dose lutein group; ALC + LUT2: alcohol + medium-dose lutein group; ALC + LUT3: alcohol + high-dose lutein group; a, vs. the control group, *p* < 0.05 (CON); b, vs. the alcohol group, *p* < 0.05 (ALC); (n = 10, mean ± SD).

**Figure 6 nutrients-14-02385-f006:**
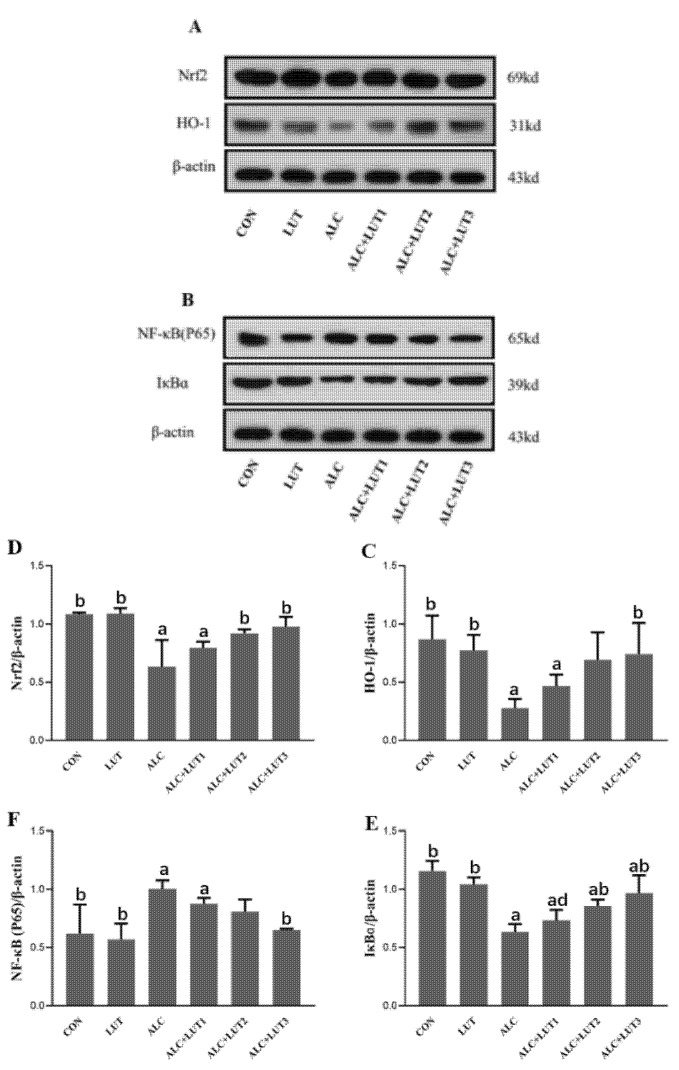
Changes in expression of proteins related to the nuclear factor kappa B (NF-κB)/nuclear factor-erythroid factor 2-related factor 2 (Nrf2) pathway. (**A**,**B**) Effects of lutein on the expression of proteins in the NF-κB and Nrf2 pathways in alcohol-treated rats. The protein levels of heme oxygenase-1 (HO-1) (**C**), Nrf2 (**D**), inhibitor of nuclear factor kappa B α (IκBα) (**E**), and NF-κB (P65) (**F**). CON: control group; LUT: lutein alone supplement group; ALC: alcohol model group; ALC + LUT1: alcohol + low-dose lutein group; ALC + LUT2: alcohol + medium-dose lutein group; ALC + LUT3: alcohol + high-dose lutein group. a, vs. the control group, *p* < 0.05 (CON); b, vs. the alcohol group, *p* < 0.05 (ALC); d, vs. the ALC + LUT3 group, *p* < 0.05. (Values are expressed as mean ± SD; n = 3).

**Figure 7 nutrients-14-02385-f007:**
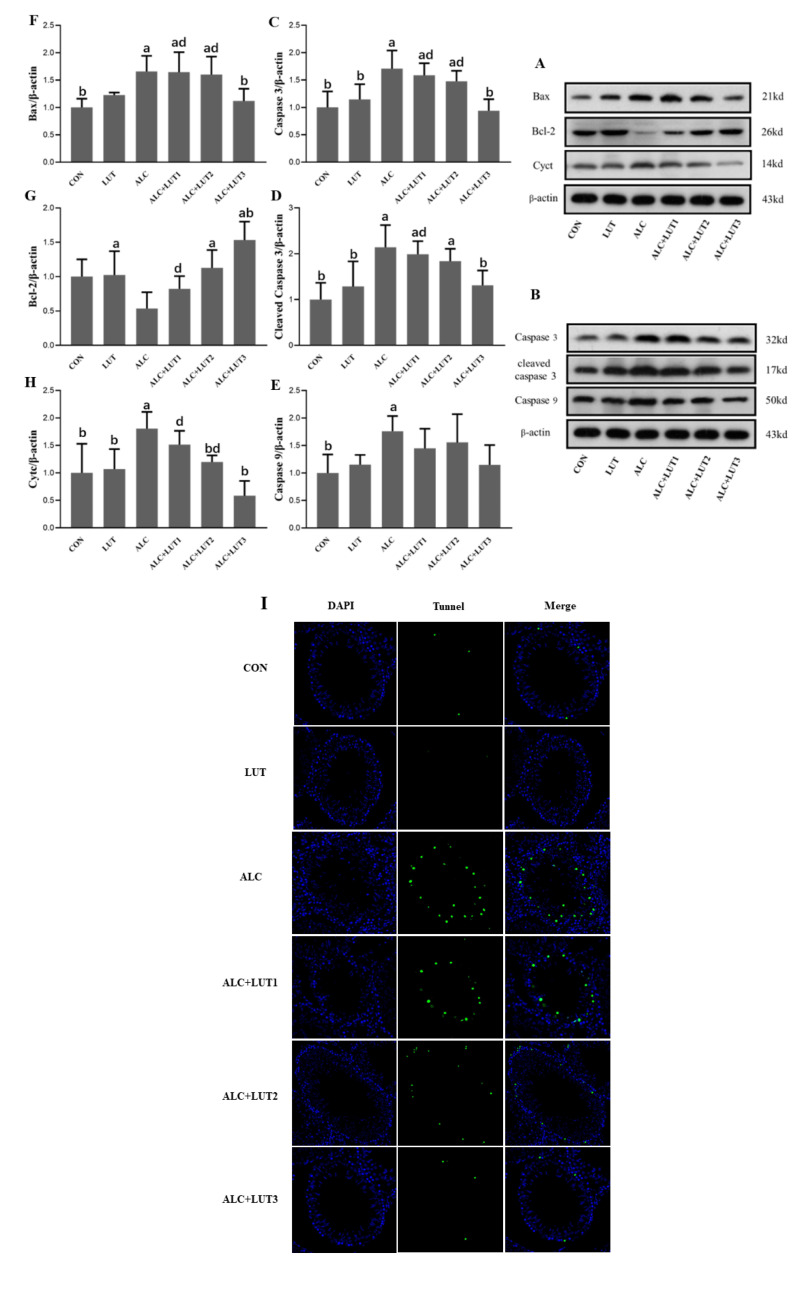
Apoptosis of testicular germ cells and expression of related proteins. (**A**,**B**) Western blot tests for apoptotic protein expression levels in testis. The protein levels of caspase-3 (**C**), cleaved caspase-3 (**D**), caspase-9 (**E**), B-cell lymphoma 2 (Bcl-2) (**F**), Bcl-2-associated X Bax (**G**), and cytosolic cytochrome c (Cytc) (**H**). Values are expressed as mean ± SD (n = 3). a, vs. the control group, *p* < 0.05 (CON); b, vs. the alcohol group, *p* < 0.05 (ALC); d, vs. the ALC + LUT3 group, *p* < 0.05. (n = 3, mean ± SD). (**I**) terminal deoxynucleotidyl transferase dUTP nick-end labeling (TUNEL) staining in rat testes. Apoptotic cells were imaged as green and nuclei were imaged as blue. CON: control group; LUT: lutein alone supplement group; ALC: alcohol model group; ALC + LUT1: alcohol + low-dose lutein group; ALC + LUT2: alcohol + medium-dose lutein group; ALC + LUT3: alcohol + high-dose lutein group.

**Table 1 nutrients-14-02385-t001:** The level of marker enzyme in the testis.

Group	LDH (U/g Protein)	AKP (U/g Protein)	ACP (U/g Protein)	SDH (U/mg Protein)
CON	396.26 ± 74.62	82.77 ± 20.83 ^b^	50.33 ± 6.65 ^b^	5.11 ± 0.88 ^b^
LUT	409.00 + 115.56	71.82 ± 19.86	46.08 ± 5.63	4.56 ± 0.88
ALC	366.91 ± 114.79	61.10 ± 23.12 ^a^	39.90 ± 6.75 ^a^	3.99 ± 0.65 ^a^
ALC + LUT1	344.48 ± 155.26	58.37 ± 20.53 ^ad^	41.42 ± 4.54 ^a^	4.26 ± 0.61 ^a^
ALC + LUT2	382.60 ± 176.02	70.51 ± 30.88	43.32 ± 7.41 ^a^	4.08 ± 0.55 ^a^
ALC + LUT3	391.11 ± 141.63	86.01 ± 16.73 ^b^	45.32 ± 10.77	4.65 ± 0.38 ^b^

Note: LDH, lactate dehydrogenase; AKP, alkaline phosphatase; ACP, acid phosphatase; SDH, synchronous digital hierarchy; CON: control group; LUT: lutein alone supplement group; ALC: alcohol model group; ALC + LUT1: alcohol + low-dose lutein group; ALC + LUT2: alcohol + medium-dose lutein group; ALC + LUT3: alcohol + high-dose lutein group; a, vs. the control group, *p* < 0.05 (CON); b, vs. the alcohol group, *p* < 0.05 (ALC); d, vs. the ALC + LUT3 group, *p* < 0.05. (n = 10, mean ± SD).

**Table 2 nutrients-14-02385-t002:** Changes of oxidative stress level in testicular tissue.

Group	SOD (U/mg Protein)	MDA (nmol/mL)	GSH-Px (U/mg Protein)
CON	153.58 ± 28.73	3.00 ± 1.35 ^b^	11.14 ± 1.43 ^b^
LUT	146.00 + 15.99	3.11 ± 0.70 ^b^	9.98 ± 2.48
ALC	136.66 ± 21.89	4.71 ± 1.85 ^a^	8.25 ± 2.70 ^a^
ALC + LUT1	139.63 ± 29.63	3.57 ± 1.30	8.41 ± 1.69 ^a^
ALC + LUT2	136.51 ± 27.34	3.70 ± 1.53	9.05 ± 2.30 ^a^
ALC + LUT3	144.75 ± 16.81	2.70 ± 0.94 ^b^	9.73 ± 1.90

Note: SOD, superoxide dismutase; MDA, malondialdehyde; GSH-Px, glutathione peroxidase; CON: control group; LUT: lutein alone supplement group; ALC: alcohol model group; ALC + LUT1: alcohol + low-dose lutein group; ALC + LUT2: alcohol + medium-dose lutein group; ALC + LUT3: alcohol + high-dose lutein group; a, vs. the control group, *p* < 0.05 (CON); b, vs. the alcohol group, *p* < 0.05 (ALC); (n = 10, mean ± SD).

## Data Availability

Not applicable.

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
