# Peer review of "Lutein Can Alleviate Oxidative Stress, Inflammation, and Apoptosis Induced by Excessive Alcohol to Ameliorate Reproductive Damage in Male Rats"

_nutrients, 2022, doi:10.3390/nu14122385_

Round 1

Reviewer 1 Report

The authors evaluated the antioxidant, anti-inflammatory and anti-apoptotic potential of lutein in testicular tissue of rats that ingested alcohol for 12 weeks. The results demonstrated lutein's potential to fight oxidative stress and inflammation promoted by excessive alcohol use. The study design is valid and presents interesting results. However, some modifications can be made to improve the final result.

Major corrections:

Transcription factors Nrf2 and NF-kb were measured by western blot, however, this technique is not the most suitable for transcription factors, as an increase in protein expression does not necessarily reflect the protein-DNA interaction. Techniques such as electrophoretic mobility shift assay, chromatin precipitation and luciferase, for example, reflect a more reliable response. If possible, confirm the results with one of them.

Minor corrections:

  1. Materials and Methods

2.2: Add project acceptance protocol number issued by Animal Care and Use Committee of the medical college, Qingdao University.

2.2: Specify in the materials and methods that concentrations of 6, 8, 10 and 12 ml/kg.bw of alcohol were given for a period of one week each. Witch alcohol? Ethanol?

  1. Results

In tables and figures, I suggest that the letter “a” of the statistic be added to the highest values, while the letter “b” to the lowest values. This standardization helps in reading the data.

3.3: In figure 3, add the group acronym in the corner of each image to facilitate visualization (similar to figure 2).

Author Response

  1. Comments to the Author

“Transcription factors Nrf2 and NF-κB were measured by western blot, however, this technique is not the most suitable for transcription factors, as an increase in protein expression does not necessarily reflect the protein-DNA interaction. Techniques such as electrophoretic mobility shift assay, chromatin precipitation and luciferase, for example, reflect a more reliable response. If possible, confirm the results with one of them.”

Dear reviewer: Thanks for the comment. We referred the relevant literature that the expression levels of Nrf2 and NF-κB were directly measured by western blotting [1-3]. In addition, the experiment was completed in June last year, the testicular tissue has been stored for a year, there may be a large deviation in the results. Thanks for your suggestions and guidance. The detection methods of related indicators will be impro ved in future experiments.

References:

[1] Wang Y, Zhang Z, Guo W, Sun W, Miao X, Wu H, Cong X, Wintergerst KA, Kong X, Cai L. Sulforaphane reduction of testicular apoptotic cell death in diabetic mice is associated with the upregulation of Nrf2 expression and function. Am J Physiol Endocrinol Metab. 2014 Jul 1;307(1): E14-23.

[2] Lin YS, Liu CY, Chen PW, Wang CY, Chen HC, Tsao CW. Coenzyme Q amends testicular function and spermatogenesis in male mice exposed to cigarette smoke by modulating oxidative stress and inflammation. Am J Transl Res. 2021 Sep 15;13(9):10142-10154.

[3] Singh S, Singh SK. Prepubertal exposure to perfluorononanoic acid interferes with spermatogenesis and steroidogenesis in male mice. Ecotoxicol Environ Saf. 2019 Apr 15;170: 590-599.

  1. Comments to the Author

“Add project acceptance protocol number issued by Animal Care and Use Committee of the medical college, Qingdao University.”

Dear reviewer: Thanks for the comment. The issued number is No.20210315Wistar900706. In the revised manuscript, the content has been added (Page 2, Line 95) as below:

All animal experiments were strictly abided by the Guidelines for the Care and Use of Laboratory Animals of the National Institutes of Health, and were approved by the Animal Care and Use Committee of the Medical College (No.20210315Wistar900706), Qingdao University.”

  1. Comments to the Author

“Specify in the materials and methods that concentrations of 6, 8, 10 and 12 ml/kg.bw of alcohol were given for a period of one week each. Witch alcohol? Ethanol?”

Dear reviewer: Thanks for the comment. According to your suggestion, we have described the alcohol used in detail in the article. In the revised manuscript, the content has been added (Page 2, Line 73-74; Page 3, Line 98-101) as below:

Alcohol (56% (v/v) ethanol) was purchased from Beijing Hongxing Alcohol Co., Ltd. (Beijing, China).”

alcohol model group (ALC), 6, 8, 10 and 12mL/(kg.bw.d) of 56% (v/v) ethanol was given in order to make rats adapt to alcohol intake. After the body weight of the rats was stable, 12mL/(kg.bw.d) of 56% (v/v) ethanol and the same amount of corn oil were given every day.

In this study, we preliminarily discussed the alcohol concentration. After 12mL/ (kg.bw.d) ethanol intake, the alcohol concentration (68.36mg/100mL) in rats was closest to the normal human alcohol concentration (79.97mg/100mL), and the mortality was low. The method of intragastric administration with gradually increasing alcohol concentration was used to make rats adapt to alcohol intake.

  1. Comments to the Author

“In tables and figures, I suggest that the letter “a” of the statistic be added to the highest values, while the letter “b” to the lowest values. This standardization helps in reading the data.”

Dear reviewer: Thanks for your suggestion. The tables and Figures were modified as suggested (Page 7, Line 245; Page 8, Line 273).

5.Comments to the Author

“In figure 3, add the group acronym in the corner of each image to facilitate visualization (similar to figure 2).”

Dear reviewer: Thanks for your suggestion. The Figure 3 was modified as suggested (Page 7, Line 225).

Reviewer 2 Report

Regarding manuscript entitled “Lutein can alleviate oxidative stress, inflammation and apopto- 2 sis induced by excessive alcohol to improve male reproductive 3 damage”, the authors aimed to confirm the alleviating effect of lutein on alcohol-induced reproductive toxicity and to evaluate the possible potential mechanisms in the in vivo system with the view of a possible treatment for testicular injury induced by alcohol.

The study is interesting and well designed. However, major revision is needed before its acceptance.

Abstract needs great improvement. Add the aim of study and the important points of material and methods to abstract.

The authors must use One-way ANOVA to compare all 6 groups at the same time. For example, it is important to compare between ALC+LUT1, ALC+LUT2 and ALC+LUT3 groups to know the best dose of lutein. The authors can use letters a-f to show significant differences between 6 groups. The results and discussion must be revised after new statistical comparison.  

Figures 2, photos of sperm slides add no value.

All abbreviations must be defined in figures and tables including groups and measured parameters.

Author Response

Response to comments from the Editors and Reviewers

Dear Editors and Reviewers,

Thank you for taking time to review our manuscript and giving valuable comments. We responded to the comments point-by-point, and revised our manuscript accordingly. In the response below, the comments were written in black, and our responses were written in blue.

Academic Editor:

  1. Comments to the Author

“I propose to change of the title of manuscript. The present title version suggests that experiments were performed on humans.”

Dear editor: Thanks for the comment. We have modified the title. In the revised manuscript, the content has been modified (Page 1, Line 2-4) as below:

Lutein can alleviate oxidative stress, inflammation, and apoptosis induced by excessive alcohol to ameliorate reproductive damage in male rats

  1. Comments to the Author

“Conclusions - the term "alcoholism" sholud be changed for 'chronic alcohol poisoning".

Dear editor: Thanks for the comment. We have modified the content. In the revised manuscript, the content has been modified (Page 13, Line 457-458, Line 465) as below:

“As is known, this is the first study of the protective effect of lutein on male repro-ductive damage caused by chronic alcohol poisoning.”

“Lutein supplementation can be considered as an adjuvant therapy for the prevention of reproductive dysfunction caused by chronic alcohol poisoning.”

Reviewer: 2

  1. Comments to the Author

“Abstract needs great improvement. Add the aim of study and the important points of material and methods to abstract.”

Dear reviewer: Thanks for the comment. The aims and material and materials were added in the abstract as suggested (Page 1, Line 10-24), as below.

“Chronic excessive alcohol intake may lead to male reproductive damage. Lutein is a carotenoid compound with antioxidant activity. The purpose of this study was to observe the effect of lute-in supplementation on male reproductive damage caused by excessive alcohol intake. In this study, an animal model of excessive drinking (12 mL/(kg.bw.d)) for 12 weeks was established and supplemented with different doses of lutein (12, 24, 48 mg/(kg.bw.d)). The results showed that the body weight, sperm quality, sex hormones (FSH, testosterone) and antioxidant markers (GSH-Px) decreased significantly, while MDA and inflammatory factors (IL-6, TNF- α) increased significantly in alcohol model group than in the normal control group. After 12 weeks of high-dose lutein supplementation with 48mg/(kg.bw.d), the spermatogenic ability, testosterone level, and the activity of marker enzymes reflecting testicular injury were improved. In addition, high-dose lutein supplementation downregulated the NF-κB and the pro-apoptosis biomarkers (Bax, Cytc and caspase-3), whereas it upregulated the expression of Nrf2/HO-1 and the anti-apoptotic molecule Bcl-2. These findings were fully supported by analyzing testicular histopathology and measuring germ cell apoptosis. In conclusion, lutein protects against reproductive injury in-duced by excessive alcohol by its antioxidant, anti-inflammatory, and anti-apoptotic properties.”

  1. Comments to the Author

“The authors must use One-way ANOVA to compare all 6 groups at the same time. For example, it is important to compare between ALC+LUT1, ALC+LUT2 and ALC+LUT3 groups to know the best dose of lutein. The authors can use letters a-f to show significant differences between 6 groups. The results and discussion must be revised after new statistical comparison.”

Dear reviewer: Thanks for the comment. We have modified the relevant content, and have explained the differences between different doses of lutein supplementation groups in the results (Page 4, Line 173-176; Page 5, Line 201-204; Page 7, Line 241-242; Page 7, Line 255-256; Page 9, Line 304-306; Page 9, Line 322-324) and elaborated the optimal dose groups in the discussion.

  1. Comments to the Author

“Figures 2, photos of sperm slides add no value.”

Dear reviewer: Thanks for the comment. Considering that sperm morphology can directly observe the occurrence of sperm malformation, and related literatures [1-2] generally use this result to evaluate sperm quality. We have also made corresponding improvements to this part of the results (Page 6, Line 225), and hope to consider retaining this part of the results.

References:

[1] Hou B, Wang F, Liu T, Wang Z. Reproductive toxicity of polystyrene microplastics: In vivo experimental study on testicular toxicity in mice. J Hazard Mater. 2021 Mar 5;405: 124028.

[2] Zhou L, Zhang C, Qiang Y, Huang M, Ren X, Li Y, Shao J, Xu L. Anthocyanin from purple sweet potato attenuates lead-induced reproductive toxicity mediated by JNK signaling pathway in male mice. Ecotoxicol Environ Saf. 2021 Aug 24; 224: 112683.

  1. Comments to the Author

“All abbreviations must be defined in figures and tables including groups and measured parameters.”

Dear reviewer: Thanks for your suggestion. The tables and figures were modified as suggested.

Reviewer 3 Report

Dear authors! I have the following comments on the your  article manuscript.

  1. In my opinion, the authors did not sufficiently substantiate the relevance of this study. Thus, the Introduction needs to be improved.
  2. Figure 2 needs improvement. It is necessary to add the missing inscriptions A-D to the figures. Figures 2A and 2B visualize very large values of the statistical error of the mean (SEM). This casts doubt on the significance of the differences between the variants.
  3. Section 3.4 contains a lot of enzyme abbreviations that need to be deciphered. Moreover, the need to evaluate these enzymes should be substantiated.
  4. Columns 2 and 3 of Table 1 contain data that do not seem adequate. The statistical error of the mean in these data reaches 45 percent of the mean. Under these conditions, it is impossible to consider the results statistically significant.
  5. Line 377 «lutein- 377 treated rats decreased the levels of inflammatory cytokines (IL-6 and TNF- α)» — This statement is not supported by experimental data in full (Figure 5). Fixes needed.
  6. Line 407 «As far as I know» — The authors allow expressions that are unacceptable in a scientific text.

Thus, the manuscript requires serious changes and evidence that the conclusions are justified.

Author Response

  1. Comments to the Author

“In my opinion, the authors did not sufficiently substantiate the relevance of this study. Thus, the Introduction needs to be improved.”

Dear reviewer: Thanks for the comment. The introduction has been modified and explained the relevance of the study (Page 2, Line 28-68).

  1. Comments to the Author

“Figure 2 needs improvement. It is necessary to add the missing inscriptions A-D to the figures. Figures 2A and 2B visualize very large values of the statistical error of the mean (SEM). This casts doubt on the significance of the differences between the variants.”

Dear reviewer: Thanks for the comment. We have modified the picture and re-examined the data of sperm count and sperm motility, and found that the group with too large standard deviation had a high coefficient of variation and a large extreme value. We deleted the maximum value of sperm count in CON group and ALC+LUT3 group, and sperm motility rate in ALC+LUT1 group, and the standard deviation was improved. However, the differences among the groups have also been changed and have been revised in this paper.

(1) The sperm count of the CON group before the maximum value was deleted, that is, the data in the original text, its mean ±SD was 39.20 ±16.07, and the coefficient of variation (CV) was 40.99%. After deleting the maximum value of CON group, mean ± SD was 36.44 ± 6.91 and CV was 18.96%.

(2) Before the maximum value of sperm count was deleted in the ALC+LUT3 group, that is, the data in the original text, its mean ± SD was 36.50 ± 19.19, and the coefficient of variation (CV) was 52.57%. After deleting the maximum of this group, the mean ± SD was 30.77 ± 6.77, and the coefficient of variation (CV) was 22%.

The Mean ± SD of sperm motility rate of ALC+LUT1 group was 33.92 ± 17.89 and coefficient of variation (CV) was 52.74% before deleting the maximum value, and the Mean ± SD was 31.11 ± 7.41 and CV was 23.81% after deletion of the maximum value.

  1. Comments to the Author

“Section 3.4 contains a lot of enzyme abbreviations that need to be deciphered. Moreover, the need to evaluate these enzymes should be substantiated.”

Dear reviewer: Thanks for the comment. The abbreviations of testicular marker enzymes were annotated in Table 1. The significance and importance of testicular marker enzymes were described in detail in the discussion section (Page 11, Line 364-371, Line 372-376), as below:

“Testicular marker enzymes can be used to evaluate the severity of testicular tissue in-jury, and also affect the formation and maturation of spermatozoa [16]. The marker enzyme ACP, located in Sertoli cells, scavenges damaged or senescent cells and maintains normal metabolism of testis. AKP is related to cell proliferation, metabolism, energy transfer, and other activities [16]. SDH is an important enzyme in the mitochondrial Kreb cycle, mainly related to the aerobic oxidation of acetyl-CoA and the production of ATP [17], affecting the development of germ cells [18].”

“Some studies have suggested that the disorder of marker enzymes caused by alcohol loading may be due to the strong cytotoxicity of mitochondrial lipid peroxidation in the testis, resulting in the disorder of energy metabolism, oxidative phosphorylation, Kreb cycle, and glycolysis. However, the disorder of marker enzymes can further affect the production of nutrients needed to support sperm maturation [19].”

Referring to the previous related research [1-2], the marker enzyme can reflect the degree of testicular tissue damage caused by alcohol. In addition, marker enzymes can provide energy and nutrients for sperm production. Lutein can regulate the level of testicular marker enzymes by reducing oxidative stress caused by alcohol, so as to maintain normal spermatogenesis.

References:

[1] Song Y, Jia ZC, Chen JY, Hu JX, Zhang LS. Toxic effects of atrazine on reproductive system of male rats. Biomed Environ Sci. 2014 Apr;27(4):281-8.

[2] Sai L, Li X, Liu Y, Guo Q, Xie L, Yu G, Bo C, Zhang Z, Li L. Effects of chlorpyrifos on reproductive toxicology of male rats. Environ Toxicol. 2014 Sep;29(9):1083-8.

  1. Comments to the Author

“Columns 2 and 3 of Table 1 contain data that do not seem adequate. The statistical error of the mean in these data reaches 45 percent of the mean. Under these conditions, it is impossible to consider the results statistically significant.”

Dear reviewer: We reanalyzed the data in the second and third columns of Table 1 (i.e., LDH and AKP levels), and found that the data whose standard deviation reached 45% of the mean value were mainly LDH levels in ALC+LUT1 and ALC+LUT2 groups, and AKP levels in ALC+LUT2 groups. However, no statistical difference was found in these data. Considering the limited number of rats in each group (only 10), the stability of the results would be affected to some extent. In future experiments, the number of rats could be increased to reduce the differences within the group as much as possible.

  1. Comments to the Author

“Line 377 «lutein- 377 treated rats decreased the levels of inflammatory cytokines (IL-6 and TNF-α) » — This statement is not supported by experimental data in full (Figure 5). Fixes needed.”

Dear reviewer: Thanks for the comment. The statement was modified as suggested (Page 13, Line 427-428), as below:

“However, rats in the high-dose lutein supplement group had lower levels of inflammatory cytokines (IL-6 and TNF-α) and expression of NF-κB than those in the alcohol-treated group.”

  1. Comments to the Author

“Line 407 «As far as I know» — The authors allow expressions that are unacceptable in a scientific text.”

Dear reviewer: Thanks for the comment. The expression was modified as suggested (Page 13, Line 457), as below:

“As is known, this is the first study of the protective effect of lutein on male reproductive damage caused by chronic alcohol poisoning.”

Round 2

Reviewer 2 Report

The authors did not respond to the following comments:

“The authors must use One-way ANOVA to compare all 6 groups at the same time. For example, it is important to compare between ALC+LUT1, ALC+LUT2 and ALC+LUT3 groups to know the best dose of lutein. The authors can use letters a-f to show significant differences between 6 groups. The results and discussion must be revised after new statistical comparison.”

“Figures 2, photos of sperm slides add no value.”

Author Response

  1. Comments to the Author

“The authors must use One-way ANOVA to compare all 6 groups at the same time. For example, it is important to compare between ALC+LUT1, ALC+LUT2 and ALC+LUT3 groups to know the best dose of lutein. The authors can use letters a-f to show significant differences between 6 groups. The results and discussion must be revised after new statistical comparison.”

Dear reviewer: Thanks for the comment. We have modified the relevant content, and have explained the differences between different doses of lutein supplementation groups in the results and elaborated the optimal dose groups in the discussion. (a: compared with the control group; b: compared with the alcohol model group; c: compared with the middle dose lutein group; d: compared with the high-dose lutein group, there was significant difference),

Page 4, Line 169-172:“In addition, the testis and epididymis weights in the ALC+LUT1 group were significantly lower those in the ALC+LUT2 and ALC+LUT3 groups, and the epididymal coefficient in the ALC+LUT1 group was significantly lower than that in the ALC+LUT2 group (P < 0.05).”

Page 5, Line 197-200:In addition, compared with ALC+LUT3 group, the sperm deformity rate of the ALC+LUT1 group and ALC+LUT2 group was significantly higher than that of the high dose group by 37.85% and 15.10%, respectively (P < 0.05). indicating that the supplement of high-dose lutein was more effective in improving the rate of sperm deformity.

Page 6, Line 237-238:“The activity of AKP in the ALC+LUT1 group was 32.13% lower than that in the ALC+LUT3 group, and the difference was statistically significant (P < 0.05).”

Page 7, Line 250-251:The testosterone level of ALC+LUT1 group was significantly lower than that of ALC+LUT3 group by 19.39% (P < 0.05).

Page 9, Line 300-302:“The expression of IκBɑ was also different in the various lutein dose groups, and the expression of lutein in the low dose lutein group was significantly lower than that in high-dose group (P < 0.05).

Page 9, Line 317-319:The expression levels of Bax, caspase-3, and cleaved caspase-3 in the ACL+LUT1 group and Bax and caspase-3 in the ACL+LUT2 group were significantly higher than those in the ACL+LUT3 group (P < 0.05).

  1. Comments to the Author

“Figures 2, photos of sperm slides add no value.”

Dear reviewer: Thanks for the comment. Considering that sperm morphology can directly observe the occurrence of sperm malformation, and related literatures [1-2] generally use this result to evaluate sperm quality. We have also made corresponding improvements to this part of the results (Page 6, Line 202), and hope to consider retaining this part of the results.

References:

[1] Hou B, Wang F, Liu T, Wang Z. Reproductive toxicity of polystyrene microplastics: In vivo experimental study on testicular toxicity in mice. J Hazard Mater. 2021 Mar 5;405: 124028.

[2] Zhou L, Zhang C, Qiang Y, Huang M, Ren X, Li Y, Shao J, Xu L. Anthocyanin from purple sweet potato attenuates lead-induced reproductive toxicity mediated by JNK signaling pathway in male mice. Ecotoxicol Environ Saf. 2021 Aug 24; 224: 112683.

Reviewer 3 Report

I am satisfied with the edits and improvements that the authors have made. The manuscript has clearly improved, although the introduction still seems boring

Author Response

  1. Comments to the Author

“I am satisfied with the edits and improvements that the authors have made. The manuscript has clearly improved, although the introduction still seems boring”

Dear reviewer: Thanks for the comment. The introduction has been improved again. I hope you find it interesting. (Page 2, Line 28-64).

“The incidence of infertility has increased in recent years. Among all infertile couples, the proportion of infertility caused by male factors accounts for about 50% [1]. A foreign study found [2] that male infertility is mainly due to the decrease in the number and motility of sperm in semen and the significant increase in the incidence of testicular diseases. The factors leading to the decline of sperm quality and testicular abnormalities include environmental pollutants, drugs, alcoholism, and smoking [1]. Among them, alcohol is the most common dietary factor that people are exposed to in daily life, which can induce serious male reproductive damage [3]. At present, the specific mechanism of alcohol-induced damage to testicular function has not been clarified, but free radicals produced in the process of alcohol metabolism and oxidative stress are considered to play a key role in its toxicity [4]. Oxidative stress is caused by an imbalance in antioxidant responses in the body, which can lead to adverse reactions in tissues and cells, such as lipid peroxidation and DNA damage, and further induce inflammation and apoptosis [5]. However, drugs for preventing and alleviating alcoholism are scarce and have serious side effects [6]. The supplementation of natural active substances in a reasonable diet to reduce the toxicity caused by alcohol has become a direction worth exploring.

Lutein is one of the most widely distributed carotenoids in fruits and vegetables, and it mainly exists in dark green leafy vegetables and egg yolk. Animals obtain lutein directly or indirectly from the diet because the body cannot synthesize lutein [7]. Lutein is a component of macular pigment, so it is often used to protect macula from photooxidation and enhance visual function [8]. In addition, lutein's unique structure, containing conjugated double bonds and hydroxyl groups, can serve an antioxidant role through providing electrons that with free radicals to produce more stable substances [9]. A study [10] on the intervention of lutein on ethanol-induced liver damage in rats indicated that the level of antioxidant enzymes in the livers of ethanol-treated rats was significantly increased with lutein treatment, thus reducing ethanol-induced hepatocyte injury. This further shows that lutein has a certain inhibitory effect on alcoholism-related toxicity; because the liver is the target organ of alcohol metabolism and can store a large amount of lutein, lutein’s protective effect in the liver may be more obvious than in other organs. However, there are few reports on whether lutein supple-mentation can alleviate the testicular damage caused by alcohol and whether the mechanism is the same as in previous studies.

Therefore, in this study, we established an animal model of reproductive damage caused by long-term (3 months) excessive drinking in male rats. In addition, different doses of lutein were supplemented to observe the effect of lutein on male reproductive damage caused by excessive alcohol. This study was designed to provide reference for further revealing the harm of alcohol and the reduction of the damage with lutein supplement.”